# Aspartic Acid in Health and Disease

**DOI:** 10.3390/nu15184023

**Published:** 2023-09-17

**Authors:** Milan Holeček

**Affiliations:** Department of Physiology, Faculty of Medicine in Hradec Králové, Charles University, Šimkova 870, 500 03 Hradec Králové, Czech Republic; holecek@lfhk.cuni.cz

**Keywords:** malate–aspartate shuttle, urea cycle, gluconeogenesis, branched-chain amino acids, neurotransmission, purine-nucleotide cycle, aspartame, oxaloacetate, aspartate and cell-to-cell interactions, glutamate–glutamine cycle

## Abstract

Aspartic acid exists in L- and D-isoforms (L-Asp and D-Asp). Most L-Asp is synthesized by mitochondrial aspartate aminotransferase from oxaloacetate and glutamate acquired by glutamine deamidation, particularly in the liver and tumor cells, and transamination of branched-chain amino acids (BCAAs), particularly in muscles. The main source of D-Asp is the racemization of L-Asp. L-Asp transported via aspartate–glutamate carrier to the cytosol is used in protein and nucleotide synthesis, gluconeogenesis, urea, and purine-nucleotide cycles, and neurotransmission and via the malate–aspartate shuttle maintains NADH delivery to mitochondria and redox balance. L-Asp released from neurons connects with the glutamate–glutamine cycle and ensures glycolysis and ammonia detoxification in astrocytes. D-Asp has a role in brain development and hypothalamus regulation. The hereditary disorders in L-Asp metabolism include citrullinemia, asparagine synthetase deficiency, Canavan disease, and dicarboxylic aminoaciduria. L-Asp plays a role in the pathogenesis of psychiatric and neurologic disorders and alterations in BCAA levels in diabetes and hyperammonemia. Further research is needed to examine the targeting of L-Asp metabolism as a strategy to fight cancer, the use of L-Asp as a dietary supplement, and the risks of increased L-Asp consumption. The role of D-Asp in the brain warrants studies on its therapeutic potential in psychiatric and neurologic disorders.

## 1. Introduction

Aspartic acid is a nutritionally non-essential amino acid discovered by hydrolysis of asparagine [1]. Aspartate exists in two isoforms; the main form is L-aspartic acid (L-Asp), and D-aspartic acid (D-Asp) is present in much smaller amounts:



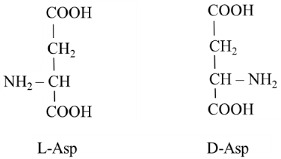



L-Asp has exceptional importance in urea synthesis, purine-nucleotide cycle (PNC), malate–aspartate shuttle (MAS), gluconeogenesis, and neurotransmission, and it is the substrate for the synthesis of proteins, asparagine, arginine, nucleotides, and of several substances that play a role in the development of nervous tissue and neurotransmission (Figure 1). D-Asp has unique significance in brain development and the regulation of hypothalamus function. 

This narrative review aims to examine the origin and pathways of aspartate metabolism in physiological and various pathological conditions, the therapeutic potential of targeting pathways of aspartate metabolism, and the use of aspartate and aspartate-containing substances as nutritional supplements. The focus is on the role of mitochondrial carriers in the compartmentalization of L-Asp and glutamate metabolic pathways between mitochondria and cytosol [2,3,4,5] and new insights into the role of aspartate in cell-to-cell interactions and neurotransmission in the brain, cell proliferation, glycolysis, gluconeogenesis, diabetes, psychiatric and neurologic disorders, liver cirrhosis, and cancer [6,7,8,9,10,11,12,13,14,15,16,17,18].

## 2. Aspartate Transporters

Aspartic acid has two negatively charged carboxyl groups at physiological pH and is transported through cell membranes via specific transporters together with glutamic acid, the other dicarboxylic amino acid found in the body. 

### 2.1. Plasma Membrane Transporters 

The major role in aspartate transport across the plasma membrane from extracellular to intracellular space has “excitatory amino acid transporters” (EAATs) 1–5 encoded by SLC1A1 (EAAT3), SLC1A2 (EAAT2), SLC1A3 (EAAT1), SLC1A6 (EAAT4), and SLC1A7 (EAAT5) [19]. The EAATs couple dicarboxylic amino acids uptake (glutamate, L-Asp, D-Asp) to co-transport three sodium ions and one proton and counter-transport one potassium ion to accumulate amino acids against up to 106-fold concentration gradients [7]. The expression is ubiquitous, especially in the brain (neurons, astrocytes, Bergmann glia, oligodendrocytes, Purkinje cells, etc.), liver, pancreas, and apical side of enterocytes, and cells of proximal tubules in the kidneys [19].

Depolarization-induced release from synaptic vesicles is the main form of aspartate release to the extracellular space in the brain. It has been shown that aspartate is co-localized together with glutamate, the main excitatory neurotransmitter, in the same nerve terminals in separate vesicle populations. The molecular mechanisms of regulation of independent exocytotic release of aspartate and glutamate have to be elucidated [6].

High L-Asp concentration in tissues (0.5–5 mM) and its very low level in the blood (~3 µM) [20,21] indicate the need for facilitated transport to allow L-Asp release from cells along its concentration gradient, particularly through the basolateral side of enterocytes and tubular cells of the kidneys. However, the existence of a specific efflux carrier for L-Asp and glutamate has not been proven. Hence, efficient L-Asp utilization in enterocytes and proximal tubules [22,23,24] suggests that the specific transporter for L-Asp efflux from the cell does not exist.

### 2.2. Mitochondrial Transporters 

#### 2.2.1. Aspartate–Glutamate Carrier (AGC)

Two types of AGC that ensure L-Asp transport from the mitochondria to the cytosol in exchange with glutamate have been identified and designated as AGC1 (SLC25A12, aralar 1) and AGC2 (SLC25A13, citrin). AGC1 is expressed in the heart, skeletal muscle, and brain, whereas AGC2 is typical for the liver and gastrointestinal tract [4,25,26,27]. The driving force for L-Asp and glutamate exchange is the proton gradient created by the respiratory chain of the mitochondria. Cells with impaired mitochondrial function have low L-Asp levels and slow proliferation [3]. 

Both carriers are stimulated by Ca^2+,^ which binds to a unique regulatory domain facing the intermembrane space [2,27]. Therefore, the signals that affect the cytosolic concentration of Ca^2+^ modulate the flux of L-Asp and glutamate through AGC1/2. These are mainly glucagon and catecholamines, which elevate cellular Ca^2+^ levels by stimulation of its release from the endoplasmic reticulum or uptake from the extracellular fluid [28,29,30].

Although AGC1 and AGC2 have the same role in the mitochondrial membrane, their physiological roles differ due to the different functions of the tissues in which they are found. Specifically, AGC1 plays a unique role in PNC, whereas AGC2 does so in the urea cycle and gluconeogenesis [5].

#### 2.2.2. Uncoupling Protein 2 (UCP2, Aspartate/Pi + H^+^ Exchanger)

Human UCP2 encoded by the SLC25A8 gene is a ubiquitously expressed mitochondrial transporter that catalyzes the export of L-Asp, malate, and oxaloacetate (OA) out of mitochondria against phosphate plus a proton. Studies have shown that UCP2 plays a role in antioxidant defense, reactive oxygen species production, insulin secretion, and tumor growth [31,32,33,34]. It has recently been discovered that UCP5 (BMCP1, brain mitochondrial carrier protein 1) encoded by SLC25A14 also has the ability to transport L-Asp [35]. 

## 3. Aspartate Origin

### 3.1. Origin of L-Asp

Under physiological conditions, most L-Asp obtained from food is transported through the apical membrane of enterocytes by SLC1A1 (EAAT3) and utilized for the synthesis of other amino acids (alanine, glutamate, proline, ornithine, and citrulline), nucleotides, and ATP [19,22,36]. It has been shown that less than 1% of L-Asp administered alone or with 18 other amino acids plus glucose was recovered intact in intestinal blood [22]. Therefore, L-Asp concentrations in the blood are very low, and the main source of L-Asp is its synthesis by aspartate aminotransferase (AST) in most tissues.

AST (L-aspartate-2-oxoglutarate aminotransferase, EC 2.6.1.1), formerly called glutamic oxaloacetic transaminase (GOT) and the well-known blood biochemical indicator of liver and heart injury, is a pyridoxal phosphate-dependent enzyme that catalyzes the interconversion of L-Asp and 2-oxoglutarate (2-OG) to oxaloacetate and glutamate (Glu). The reaction is close to equilibrium and can be readily reversed. Therefore, L-Asp can be synthesized as well as degraded depending on the reactant concentrations (L-Asp + 2-OG ↔ OA + Glu). The enzyme exists in cytoplasmic (cAST) and mitochondrial (mAST) forms [37,38]. 

Under ordinary conditions, the net flux through mAST is toward the direction of L-Asp because of a continuous supply of glutamate from the cytosol into the mitochondria by several transporters, including SLC25A12 (AGC1), SLC25A13 (AGC2), SLC25A18, and SLC25A22, whereas L-Asp is effectively exported to the cytosol via AGC1/2 and UCP2/5:OA + Glu → L-Asp + 2-OGUnlike mAST, the cAST functions mainly in the direction of glutamate and oxaloacetate formation due to the continuous supply of L-Asp from the mitochondria to the cytosol and opposite transport of glutamate by AGC1/2, SLC25A18, and SLC25A22: L-Asp + 2-OG → OA + Glu

Sources of glutamate for L-Asp synthesis in the mitochondria are tissue-dependent.

Glutamate, the donor of nitrogen for L-Asp synthesis by mAST, is obtained from the blood or synthesized during amino acid catabolism. Especially important sources of glutamate are glutamine and branched-chain amino acids (BCAAs: valine, leucine, and isoleucine) (Figure 2). 

Glutamine, which is the most abundant amino acid in extracellular fluid, is the main source of glutamate in tissues with high expression of mitochondrial glutaminase (Gln + H_2_O → Glu + NH_3_), particularly in the liver and kidneys and rapidly proliferating cells such as enterocytes and tumor cells [39,40]. Recently, novel glutaminase isoforms with extramitochondrial locations have been discovered in the brain [41]. Glutamine is transported into the cells primarily by some sodium-neutral amino acid transporters (SNAT2-4; SLC38A2-4). Across the inner mitochondrial membrane is transported by a variant of SLC1A5 [42]. 

The BCAAs are the main source of amino nitrogen for glutamate synthesis in tissues with high expression of BCAA aminotransferase (BCAA + 2-OG ↔ BCKA + Glu). The enzyme exists in two isoforms, one located in mitochondria and the other in the cytosol. In humans, mitochondrial activity is high in skeletal muscle, the colon, and kidneys, whereas the cytoplasmic isoform is found only in the brain [43]. Very low expression of the enzyme is in the liver [44]. Therefore, unlike other amino acids, the initial step of BCAA catabolism does not take place in the liver but in muscles [44,45]. The BCAAs are removed from the circulation by LAT1 (SLC7A5/SLC3A1) and used preferentially for protein synthesis. The surplus is delivered from the cytosol to the mitochondria by SLC25A44 [46] to act in a reaction catalyzed by BCAA aminotransferase as the main nitrogen donor for glutamate synthesis [44,47,48,49].

### 3.2. Origin of D-Asp

D-Asp can be obtained together with other D-amino acids from food and gut microbiota or synthesized from L-Asp using a specific racemase and degraded by specific oxidase to oxaloacetate, ammonia, and H_2_O_2_ [50]:L-Asp → D-Asp (racemase)
D-Asp + H_2_O + O_2_ → H_2_O_2_ + NH_4_^+^ + OA (oxidase)

High D-Asp levels are in the brain of the fetus and testes, suggesting its role in brain development and spermatogenesis and testosterone synthesis, respectively [51,52,53,54,55].

## 4. Metabolism and Physiologic Importance of Aspartic Acid

L-Asp synthesis in mitochondria and its transport from the mitochondria by AGC1/2 or UCP2/5 has fundamental importance for ensuring several metabolic pathways that take place in the cytosol. 

### 4.1. Aspartate and Protein Synthesis

The single-letter amino acid symbol “D” is used in protein sequences to indicate the presence of aspartate, whose global frequency is around 8% [56]. Since L-Asp is a polar amino acid, its residues are mainly on the surface of proteins, where they influence interactions among proteins and their function. Side chains of L-Asp are often hydrogen bonded to the mainchain NH group to form asx turns or asx motifs (asx means L-Asp or asparagine) that enable the folding of proteins. The asx turn consists of three amino acid residues that form a hydrogen bond from its side chain CO group to the main chain NH group. The asx motif consists of four or five amino acid residues with either L-Asp or asparagine as the first residue and is formed by two internal hydrogen bonds [57]. 

In addition to L-Asp, D-Asp has been detected in the proteins of several human tissue samples, such as the eye lens, brain, skin, bone, teeth, and aorta [58]. The presence of the residues of D-Asp, which is probably the result of its racemization during tissue aging, alters protein structure and function and can play a role in disease development. The most investigated is the cataract, a cloudy area in the lens of the eye that leads to vision impairment [57].

Both L-Asp and D-Asp residues can be dehydrated on the side chain carbonyl and form L-isoAsp and D-isoAsp, respectively [59,60]. It has been shown that isoAsp formation triggers autoimmune responses to self-proteins [61]. 

### 4.2. L-Asp and Malate–Aspartate Shuttle (MAS)

The MAS (Figure 3) consists of AGC1/2 and 2-oxoglutarate carrier (OGC) and two enzymes, malate dehydrogenase (MDH) and AST, which are found both in the mitochondrial matrix and in the cytosol [4,27]. In the cytosol, L-Asp is converted by cAST to malate and then by cMDH and NADH to malate and NAD^+^. Malate transported to the mitochondria by OGC is oxidized by mMDH and NAD^+^ to OA and NADH. The electrons of NADH are funneled into the respiratory chain to be used by ATP synthase to produce ATP; oxaloacetate can be used by mAST to reconstitute L-Asp. The driving force for the flux through MAS is the unidirectional transfer of L-Asp from the mitochondria to the cytosol through AGC1/2 (Section 2.2). 

In this way, MAS ensures the indirect transfer of NADH produced in the cytosol during glycolysis by 3-phosphoglyceraldehyde dehydrogenase. Other cytosolic NADH sources include pyruvate synthesis by lactate dehydrogenase and oxidation of ethanol to acetaldehyde by alcohol dehydrogenase (CH_3_CH_2_OH + NAD^+^ → CH_3_CHO + NADH + H^+^). MAS thus becomes necessary for glucose oxidation and ATP synthesis, maintaining the flux through the citric acid cycle and intracellular redox balance, and L-Asp synthesis and transport from the mitochondria to the cytosol. The relationship of MAS to urea formation, gluconeogenesis, and PNC will be described in the following sections.

### 4.3. Aspartate and Cell Proliferation

Several studies have shown that L-Asp is a limiting metabolite for cell growth and proliferation [15,16,17], which is mainly due to its role in maintaining intracellular redox balance via MAS and protein synthesis (see Section 4.1 and Section 4.2) and synthesis of nucleotides and asparagine (Figure 4).

#### 4.3.1. L-Asp and Nucleotide Synthesis

L-Asp provides one nitrogen atom to the purine ring and three carbons and one nitrogen to the rings of pyrimidines [62]. L-Asp thus becomes a key substrate for the synthesis of purines (adenine and guanine) and pyrimidines (cytosine, uracil, and thymine), and various nucleotides, such as AMP, cAMP, ATP, and GTP and nucleic acids, DNA and RNA. It is worth noting that the second source of nitrogen atoms in the structure of purines and pyrimidines is γ-nitrogen of glutamine [62] and that deamidation of glutamine by glutaminase yields glutamate, the direct substrate for L-Asp synthesis. Therefore, the use of L-Asp in the synthesis of nucleotides is closely associated with glutamine metabolism. 

Although the basic nucleotide synthesis pathways are known and described in detail in biochemistry textbooks [62,63], the regulatory networks for modulating nucleotide biosynthesis remain unclear [64]. Particularly important should be understanding the pathways by which oncogenes and tumor suppressors modulate nucleotide synthesis in cancer cells [65]. 

#### 4.3.2. L-Asp and Asparagine Synthesis 

Asparagine can be formed by the amidation of L-Asp in an ATP-dependent amidotransferase reaction catalyzed by asparagine synthetase (EC 6.3.5.4) present in most mammalian organs, but basal expression is relatively low in tissues other than the exocrine pancreas [66]. Increased transcription of the enzyme appears to be a component of cell response to hypoxia or nutrient deprivation [67]. Asparagine concentrations in plasma range between 50 and 80 µM, and concentrations in intracellular fluid are up to ten times higher [20,68].

The physiologic importance of asparagine:Protein synthesis—Asparagine is a proteinogenic amino acid with a polar side chain that easily forms hydrogen bonds and increases the solubility of proteins in water. Together with L-Asp, it participates in the formation of asx turns and asx motifs that enable the folding of proteins. Its content in proteins is about 4% [56]. Like L-Asp, asparagine side chains can be post-translationally modified by a variety of chemical reactions, notably deamidation, isomerization, and racemization, resulting in various alterations in the structure and function of proteins [59].Activation of ornithine decarboxylase [69]—Ornithine decarboxylase is the rate-controlling enzyme in the biosynthesis of polyamines, e.g., spermine, spermidine, and putrescine, which are important in the regulation of cell proliferation and differentiation.Intracellular asparagine exchanges with extracellular amino acids, especially serine, arginine, and histidine, to promote mTOR activation, protein and nucleotide synthesis, and cell proliferation [70].

### 4.4. L-Asp and Arginine Synthesis

L-Asp is involved in arginine synthesis in reactions catalyzed by argininosuccinate synthetase (EC 6.3.4.5) and argininosuccinate lyase (EC 4.3.2.1), which metabolize citrulline and L-Asp to arginine and fumarate:citrulline + L-Asp + ATP → argininosuccinate + AMP + PPi
argininosuccinate → arginine + fumarate

Through this pathway, high amounts of arginine are synthesized in the urea cycle. However, there is no net synthesis of arginine by the liver during physiological conditions due to its complete consumption in the urea cycle and creatine synthesis [23]. The main source of arginine for the body’s needs is the kidneys, which uptake intestine-derived citrulline from circulation [24,71]. Studies on rats have shown that enzymes of arginine synthesis are found primarily in the proximal convoluted tubules, where most of the amino acids of glomerular filtrate are reabsorbed [24]. The rate of arginine synthesis and its transport out of the tubular cells to the blood at the basolateral membrane by facilitated diffusion is ~2 g per day [23]. 

Arginine is a proteinogenic amino acid whose metabolism ultimately results in the production of a biochemically diverse range of products, including nitric oxide, urea, creatine, polyamines, proline, glutamate, agmatine, and homoarginine [72].

### 4.5. L-Asp and the Liver

L-Asp has a unique role in the urea cycle and gluconeogenesis, the metabolic pathways that are specific to the liver (Figure 5).

#### 4.5.1. L-Asp and Urea Cycle

L-Asp enters the urea cycle in equimolar amounts with ammonia via argininosuccinate in a reaction catalyzed by argininosuccinate synthetase in the cytosol. Argininosuccinate is cleaved by argininosuccinase, which retains L-Asp nitrogen in the product arginine and releases the L-Asp carbon skeleton as fumarate. Arginine is degraded by arginase to urea and ornithine, which can, via ornithine-citrulline carrier, enter the mitochondria and react with carbamoyl phosphate to start a new cycle.

Fumarate released from the urea cycle is hydrated by cytosolic fumarase to malate, which can be utilized in two ways. Firstly, malate can be transported by OGC into the mitochondria to be converted to oxaloacetate and then to L-Asp, which can be transported by AGC2 to the cytosol and re-enter the urea cycle. This pathway that interconnects the urea cycle and the CAC is called an aspartate–argininosuccinate shunt. The second option is malate conversion to oxaloacetate, which can be used for gluconeogenesis.

#### 4.5.2. L-Asp and Gluconeogenesis 

L-Asp delivered from the mitochondria to the cytosol plays a key role in the synthesis of oxaloacetate, the starting substance for glucose synthesis, in two possible ways [18]. The first option is a direct conversion of L-Asp to oxaloacetate using cAST (L-Asp + 2-OG → OA + Glu). The second is the entrance to the urea cycle to form fumarate that is released from the cycle and hydrated to malate. Under conditions of increased supply of NAD^+^ produced during gluconeogenesis by 3-phosphoglyceraldehyde dehydrogenase or lactate dehydrogenase, a portion of the malate is diverted from entering the mitochondria by OGC to be used for OA synthesis by cMDH (malate + NAD^+^ → OA + NADH + H^+^). 

Oxaloacetate enters the reactions of gluconeogenesis via phosphoenolpyruvate carboxykinase (PEPCK) activated by the increase in the glucagon/insulin ratio during starvation or by catecholamines and cortisol during exercise and stress illness [73]. It should be emphasized that L-Asp also plays a crucial role in gluconeogenesis via its role as an intermediate in alanine synthesis in muscles (next section). 

### 4.6. L-Asp and Skeletal Muscle

In muscles, L-Asp has a unique role in PNC and glutamine and alanine synthesis, the metabolic pathways in which the primary source of nitrogen for L-Asp synthesis is the BCAA (Figure 6).

#### 4.6.1. L-Asp and Purine-Nucleotide Cycle (PNC)

The PNC comprises reactions catalyzed by adenylosuccinate synthetase, adenylosuccinate lyase, and adenosine deaminase. L-Asp, which enters the cycle via adenylosuccinate synthetase, enables, via the recycling of IMP, the balance of the levels of the adenine nucleotides (ATP, ADP, and AMP). Moreover, the PNC produces fumarate, which is largely converted to malate that can, via OGC, cross the mitochondrial membrane and enter the citric acid cycle [74,75]. Via the PNC, the amino groups of L-Asp, glutamate, and BCAA can be incorporated into ATP and become a source of ammonia:
Mitochondria:BCAA + 2-OG → BCKA + Glu
Glu + OA → 2-OG + L-AspAGC1:L-Asp transport to the cytosolCytosol:L-Asp + IMP + GTP → adenylosuccinate + GDP + Pi 
adenylosuccinate → fumarate + AMP
AMP + H_2_O → IMP + NH_3_
or
AMP + ATP → ADP + ADP
ADP + creatine phosphate → ATP + creatine

#### 4.6.2. L-Asp and Synthesis of Alanine and Glutamine in Muscles

The studies using ^15^N labeling have shown that the main source of nitrogen for L-Asp, glutamate, glutamine, and alanine synthesis in muscles is BCAA catabolism in the mitochondria [44,47,48,49]. Other studies have demonstrated that the addition of L-Asp to the medium with rat epitrochlearis muscle increases L-Ala formation [76]; BCAA supply enhances L-Asp, alanine, and glutamine levels in the blood [77]; and disruption of BCAA aminotransferase in mice increases BCAA levels in plasma while L-Asp and alanine levels are decreased [78]. Hence, in muscles, L-Asp acts as a carrier of the amino group of the BCAA to ensure alanine and glutamine synthesis in a sequence of the following reactions:
Mitochondria:BCAA + 2-OG → BCKA + Glu
Glu + OA → 2-OG + L-Asp AGC1:L-Asp transport to the cytosolCytosol:L-Asp + 2-OG → OA + Glu
Glu + pyruvate → 2-OG + Alaor
Glu + NH_3_ + ATP → Gln + ADP + Pi 

Physiologically, alanine synthesis in muscles is activated during the initial phase of starvation [79]. Alanine released from muscle by ASCT1 (SLC1A4) or other carriers is taken up by the liver using the SLC38 family of sodium-neutral amino acid transporters (SNAT2-4) and used mainly for glucose synthesis. The newly synthesized glucose can be released from the liver and returned to skeletal muscle. This closes the loop known as the glucose–alanine cycle [79].

Glutamine synthesis in muscles has a crucial role in the detoxification of ammonia formed via the purine-nucleotide cycle (Section 4.6.1) and due to hyperammonemia in liver injury or urea cycle disorders [80]. Glutamine released from the muscles to the blood by LAT1 (SLC7A5), ASCT1 (SLC1A4), and SNAT3 (SLC38A3) serves as a non-toxic form of ammonia transport for the synthesis of urea in the liver and ammonia in the kidneys, as preferred respiratory fuel and substrate for the synthesis of nucleic acids in rapidly proliferating cells, such as enterocytes, immune cells, and tumor cells, and as a precursor for glucose synthesis in the liver and kidneys [81,82,83,84,85,86]. As a signaling molecule, glutamine activates the mTORC1 (mechanistic target of rapamycin complex 1) pathway and promotes cell growth [87].

### 4.7. Aspartate and Nervous System

Both isoforms of aspartic acid have a unique role in the development, metabolism, and function of the nervous system.

#### 4.7.1. Aspartate and Neurotransmission

D-Asp and L-Asp are, together with L-glutamate, classified as excitatory neurotransmitters that depolarize the postsynaptic membrane. It is generally thought that there are no pure aspartate-releasing neurons and that aspartate and glutamate are stored within the same excitatory nerve terminal and released by a calcium-dependent exocytotic mechanism [6]. Both L- and D-aspartate bind to NMDA-type L-glutamate receptors and do not activate the α-amino-3-hydroxy-5-methyl-4-isoxazolepropionate (AMPA) type of glutamate receptors [88]. EAATs, primarily EAAT2, which transport glutamate, L-Asp, and D-Asp with similar affinity, are responsible for aspartate clearing from the synaptic cleft [7,55].

L-Asp plays a modulatory role in some excitatory pathways, particularly in the hippocampus [6,51,89,90]. Several studies have demonstrated that D-Asp is capable of releasing growth hormone from the pituitary gland, modulating melatonin synthesis in the pineal gland, and activating the hypothalamic–pituitary–gonadal axis via stimulating the release of the gonadotropin-releasing hormone from the hypothalamus [51,88,89].

#### 4.7.2. D-Asp and Brain Development

D-Asp is present at high levels in the embryo brain and strongly decreases in the post-natal phase due to the onset of D-Asp oxidase [91]. Mouse models with abnormally higher D-Asp levels have evidenced that D-Asp enhances hippocampal NMDA receptor-dependent synaptic plasticity, dendritic morphology, and memory [8,92]. 

#### 4.7.3. Aspartate-Derived Substances

L- and D-Asp are substrates for the synthesis of several substances with a fundamental role in the development and function of nervous tissue. These are mainly as follows:*N-acetyl-L-aspartate (NAA)* is an abundant brain metabolite synthesized by aspartate N-acetyltransferase from L-Asp and acetyl-CoA in neurons. It has been shown that NAA is released from neurons and transported to oligodendrocytes and astrocytes by sodium-coupled high-affinity carboxylate transporter NaC3/NaDC3, where it is cleaved by aspartoacylase into L-Asp and acetate anion that can be subsequently used for synthesis of acetyl-CoA by acetyl-CoA synthase [9,93,94]. In oligodendrocytes, acetate moiety is used for the synthesis of fatty acids and steroids, which are used for the synthesis of myelin, the basic component of the white matter of the brain and spinal cord [9].*N-acetyl-L-aspartyl-L-glutamate (NAAG)* is a neurotransmitter synthesized by peptide bond formation between NAA and glutamate that modulates glutamatergic neurotransmission [9]. NAAG is degraded by a specific dipeptidase known as NAAG peptidase, glutamate carboxypeptidase II, or prostate-specific membrane antigen to glutamate and N-acetyl-L-aspartate. The enzyme is highly expressed in the prostate and used as a biomarker of prostate cancer [95]. Inhibitors of NAAG peptidase are investigated in ischemic damage and diseases of the nervous system, including schizophrenia, diabetic neuropathy, Alzheimer’s disease, and amyotrophic lateral sclerosis [10].*N-Methyl-D-aspartate (NMDA)* is synthesized by an S-adenosyl-L-methionine-dependent enzyme from D-Asp [96]. The NMDA is a specific agonist of NMDA receptors that is used widely in neurological research [51]. It has been shown that NMDA administration can induce the release of some hypothalamic and pituitary hormones [89].

#### 4.7.4. L-Asp and Cell-to-Cell Interactions

In some glial cells, the expression of mAST and AGC1/2 is negligible, and, therefore, these cells acquire L-Asp by interacting with other cells. The interaction between neurons and astrocytes is well known. Astrocytes lack mAST and AGC1 and acquire L-Asp from neurons that are provided by both mAST and AGC1 [11,13]. A similar situation is in the retina in which both mAST and AGC1 are found in photoreceptor neurons (photoreceptors), whereas glial (Müller) cells are mAST and AGC1 deficient [14,97].

Figure 7 shows that L-Asp released from nerve endings as a neurotransmitter or in the form of NAA can be removed by astrocytes and, using cAST, be converted to oxaloacetate and glutamate. Oxaloacetate is used by the cMDH reaction to form malate that enables the transport of the NADH produced in the cytosol to the mitochondria and ensures continuous glycolysis and maintaining redox balance in astrocytes. Glutamate reacts with ammonia obtained from the blood or neurons to form glutamine, which is exported from astrocytes for subsequent release to the blood or uptake by glutamatergic or GABAergic neurons. In neurons, glutamine is hydrolyzed by the cytosolic form of glutaminase to glutamate that can be stored in synaptic vesicles for subsequent use in neurotransmission or transported via AGC1 to the mitochondria for L-Asp synthesis. 

In summary, the neuron-to-astrocyte pathway of L-Asp transport ensures glycolysis and redox balance via NADH delivery from the cytosol to the mitochondria, although the complete MAS is not expressed in astrocytes and complements the well-known glutamate–glutamine cycle in which glutamate is readily formed in neurons from glutamine synthesized in astrocytes (Figure 8) [98,99,100].

## 5. Aspartic Acid and Disease

Although aspartic acid is classified as a nutritionally non-essential amino acid, meaning that it can be synthesized in sufficient quantities in the body in all physiologic and pathologic conditions, there are several disorders in which aspartate metabolism is significantly disturbed. 

### 5.1. Primary Disorders of L-Asp Metabolism

#### 5.1.1. AGC1 (Aralar 1) Deficiency (OMIM ID #612949)

The manifestations of AGC1 deficiency are mainly due to impaired glycolysis and NAA synthesis in the brain. The patients present impaired development and function of the nervous system, including epileptic encephalopathy, hypotonia, microcephaly, abnormal myelination, and decreased NAA levels. There is a variable presence of lactic acidemia, which is probably due to the inability of the MAS to transfer NADH across the mitochondrial membrane [101]. The symptoms can be ameliorated by the ketogenic diet [102].

#### 5.1.2. AGC2 (Citrin) Deficiency (OMIM ID #603471 and #605814)

AGC2 deficiency, also called citrullinemia type 2, is an autosomal recessive disease that can manifest in newborns as “neonatal intrahepatic cholestasis”, in older children as “failure to thrive and dyslipidemia”, and in adults as “recurrent hyperammonemia” [103]. The major biochemical abnormalities include increased NADH/NAD^+^ ratio, decreased arginosuccinate synthetase activity, and low L-Asp, malate, and oxaloacetate levels in the liver. The alterations in plasma include hypercitrullinemia, hyperammonemia, hypoproteinemia, hypoglycemia, galactosemia, hyperlipidemia, increased lactate/pyruvate ratio, and decreased concentrations of L-serine. Frequent manifestations include growth failure, hepatopathy, cholestasis, and aversion to high-carbohydrate foods [101,104,105,106]. Growing evidence suggests a relationship between AGC2 deficiency and hepatocellular carcinoma [107,108,109,110,111,112].

#### 5.1.3. Asparagine Synthetase Deficiency (OMIM ID #615574)

Thus far, 15 unique mutations in the asparagine synthetase gene have been described [113]. The disease is characterized by microcephaly, disturbed myelination, severe development delay, refractory epilepsy, and death at an early age [114,115]. Asparagine supplementation is ineffective or worsens the symptoms of the disease [115,116]. 

#### 5.1.4. Canavan Disease (OMIM ID #271900)

Canavan disease is a fatal disease due to a mutation in the gene for aspartoacylase, the enzyme that breaks down NAA into L-Asp and acetate moiety. There is an accumulation of NAA and disruption in the formation of the myelin sheath of nerve fibers in the brain. The disease is inherited in an autosomal recessive manner [9].

#### 5.1.5. Citrullinemia Type 1 (OMIM ID #215700)

Citrullinemia type 1 is caused by mutations in the gene that encodes argininosuccinate synthetase that catalyzes the formation of argininosuccinate from citrulline and L-Asp. The patients exhibit hyperammonemia, hyper citrullinemia, arginine deficiency, loss of appetite, severe mental retardation, and convulsions that may progress to coma and lead to death. The main goal of the therapy is to prevent hyperammonemia using a low-protein diet and enhance glutamine excretion by phenylbutyrate or sodium benzoate [117]. 

#### 5.1.6. Dicarboxylic Aminoaciduria (OMIM ID #222730)

Dicarboxylic aminoaciduria is a rare autosomal recessive disorder due to a mutation of SLC1A1 that encodes EAAT3 found in the intestine, kidney, and brain. Patients exhibit increased loss of L-Asp and glutamate owing to impaired uptake of these amino acids in the proximal tubules [118]. The disease is mostly asymptomatic but may be linked to mental retardation, kidney stones, and schizophrenia [119]. 

### 5.2. Aspartate and Cancer

Several studies have shown that L-Asp is essential for cell proliferation (Section 4.3.) and a limiting metabolite for cancer cell growth [15,16,17]. In tumor cells, most L-Asp is derived from mitochondrial glutamine metabolism [120]. In addition to increased expression of glutamine transporters and phosphate-dependent glutaminase in cancer cells, frequently upregulated are cAST, AGC1, and asparagine synthetase [34,121,122]: *cAST*—The upregulation of cAST has been observed in pancreatic adenocarcinoma, colorectal cancer, breast cancer, acute myeloid leukemia, and hepatocellular carcinoma [121]. Although several cAST inhibitors have been developed and investigated as potential anti-cancer drugs, none has entered clinical trial [122].*AGC1*—Increased AGC1 (aralar 1, SLC25A12) expression and its mRNA levels are often elevated in tumors of the breast, pancreas, esophagus, colon, and ovaries [17,120,123,124,125]. It has been suggested that the expression of AGC1 plays a role in an increased incidence of cancer in patients with AGC2 deficiency [111,112]. SLC25A12 silencing by small interfering RNA significantly impaired HepG2 cell proliferation [111].*Asparagine synthetase*—It has been demonstrated that asparagine synthetase expression correlates with tumor grade and poor prognosis, and tumors that exhibit low asparagine synthetase activity and acquire asparagine from the extracellular environment are sensitive to therapy by asparaginase [37,67,70,126,127]. Currently, the therapy using asparaginase that depletes circulating asparagine is an integral component of chemotherapy for children and young adults with acute lymphoblastic leukemia [128]. Low expression of asparagine synthetase has also been demonstrated in human pancreatic carcinomas, suggesting the possible therapeutic potential of asparaginase therapy [129].

### 5.3. Aspartate and Psychiatric and Neurologic Disorders

In line with growing evidence of the importance of aspartate for the development, metabolism, and function of the brain, altered concentrations of aspartate and its derivatives have been reported and suggested to play a role in the development of psychiatric and neurologic disorders. These are mainly D-aspartate and NAA. 

It is supposed that the role of D-aspartate is connected mainly to its role in glutamatergic neurotransmission via NMDA receptors [130]. Decreased D-aspartate levels have been reported post-mortem in patients with schizophrenia in the prefrontal cortex and striatum [131]. On the other side, D-Asp accumulates in the brain in Alzheimer’s disease [58].

The role of NAA is probably connected to its involvement in osmoregulation, myelin synthesis, neuron–glia signaling, and turnover of NAAG and glutamate [9]. A reduction in NAA concentrations in the brain has been demonstrated in affective disorders, obsessive-compulsive disorder, schizophrenia, dementia, epilepsy, AGC1 mutations, and maple syrup urine disease, a genetic disorder caused by the deficiency of branched-chain alpha-keto acid dehydrogenase, the key enzyme in BCAA catabolism [132,133]. 

### 5.4. L-Asp and BCAA in Diabetes

In diabetes, L-Asp synthesis via mAST reaction (OA + Glu → L-Asp + 2-OG) decreases due to the decreased supply of oxaloacetate. The main cause of oxaloacetate deficiency is increased NADH production from beta-oxidation that impairs the flux through mMDH (Mal + NAD^+^ → OA + NADH + H^+^). The surplus of NADH due to enhanced fatty acid oxidation also impairs the flux through the CAC and 2-oxoglutarate production via isocitrate dehydrogenase (isocitrate + NAD^+^ → 2-OG + NADH +H^+^ + CO_2_). Since the BCAA aminotransferase reaction (BCAA + 2-OG ↔ BCKA + Glu) is reversible and near equilibrium, the surplus of glutamate (due to its decreased consumption by mAST) and 2-oxoglutarate depletion (due to decreased supply from mAST and isocitrate dehydrogenase reactions) impair the BCAA catabolism [134,135]. Increased BCAA and decreased L-Asp and oxaloacetate levels have been reported in soleus muscle in rats with diabetes induced by streptozotocin [136]. 

The increase in BCAA is the most frequent alteration in plasma amino acid levels in diabetes, which contributes to increased levels of aromatic amino acids (phenylalanine, tyrosine, and tryptophan), insulin resistance, and accumulation of various metabolites, whose influence on diabetes progression is not clear [137,138,139]. 

### 5.5. L-Asp and Hyperammonemia 

Ammonia detoxification to glutamine by glutamine synthetase in muscles is enhanced in various forms of hyperammonemia, especially liver cirrhosis and urea-cycle disorders [140,141,142]. Most of the glutamate used to detoxify ammonia by glutamine synthetase in cytosol originates from L-Asp synthesized in the mitochondria and then delivered to the cytosol by AGC1:
Mitochondria:BCAA + 2-OG → BCKA + Glu
Glu + OA → 2-OG + L-Asp AGC1:L-Asp transport to the cytosolCytosol:L-Asp + 2-OG → OA + Glu
Glu + NH_3_ + ATP → Gln + ADP + Pi 

In hyperammonemia, increased demands for L-Asp supply to form glutamate via cAST are ensured by increased flux through BCAA aminotransferase, mAST, and AGC1. The result is the loss of the BCAA, the drain of oxaloacetate and 2-oxoglutarate from the citric cycle (cataplerosis), and mitochondrial dysfunction [135,140,141,142,143,144]. It is a consensus that these alterations play a role in hepatic encephalopathy, protein-energy wasting, and increased morbidity and mortality in patients with liver cirrhosis [145,146,147].

## 6. Aspartate as a Dietary Supplement

### 6.1. L-Aspartate

Recommendations of L-Asp as a dietary supplement are based on its importance for mitochondrial function, the stimulating effect on PNC and alanine and glutamine synthesis in muscles, urea and glucose formation in the liver, and cell proliferation and its close relationship to the metabolism of glutamate, glutamine, proline, arginine, ornithine, and citrulline (Figure 9). 

#### 6.1.1. L-Asp as an Ergogenic Supplement

L-Asp is recognized as an ergogenic food supplement, and its administration (1.5–10 g before exercise) has been recommended since the 1960s [148]. Beneficial effects should be mediated by its role in MAS (delivery of NADH produced during glycolysis into mitochondria and maintaining redox balance), the stimulating effect on PNC and alanine and glutamine synthesis in muscles, and urea and glucose formation in the liver. The rationality of L-Asp supplementation is also evidenced by decreased L-Asp levels in plasma and muscles after exhausting exercise [149,150,151]. Unfortunately, the results of the studies examining the ergogenic effects of L-Asp are not convincing [148,152]. The cause may be a limited increase in the level of L-Asp in the peripheral circulation as a result of its utilization by enterocytes [22,36]. It seems promising to combine L-Asp with minerals, amino acids (e.g., ornithine, arginine, and asparagine), and other substances, such as carnitine [153,154].

#### 6.1.2. L-Asp as Ammonia Decreasing Supplement

L-Asp supply can decrease ammonia levels in two ways. First, L-Asp can, via cAST, increase the formation of glutamate (L-Asp + 2-OG → OA + Glu), which is a substrate for ammonia detoxification to glutamine in muscles. The second, L-Asp, can increase the flux through the urea cycle and enhance ammonia detoxification in the liver. The ammonia-lowering substance employed in the treatment of hepatic encephalopathy is LOLA (L-ornithine L-aspartate) [155].

#### 6.1.3. Other Possible Indications 

It has been reported that L-Asp administration may improve intestinal integrity after endotoxin challenge in weanling piglets [36], has a protective effect on cardiac function during myocardial infarction and isoproterenol-induced cardiac toxicity in rats [156,157], attenuates hypertension in Dahl salt-sensitive rats by altering arginine and NO synthesis [158], and ameliorates symptoms of diabetic kidney disease in mice [159]. 

### 6.2. Aspartate-Containing Peptides as Artificial Sweeteners

There are two peptides containing L-Asp with high affinity to gustatory receptors for a sweet taste that is used commercially as a low-calorie artificial sweetener. 

#### 6.2.1. Aspartame 

Aspartame, a dipeptide of L-Asp and phenylalanine (N-L-α-aspartyl-L-phenylalanine 1-methyl ester), is an artificial sweetener ~200 times as sweet as sucrose [160]. Under the designation E951, it is used as a low-calorie sweetener in foods, beverages, and products intended for diabetics. Its use in food processing is limited because aspartame decomposes to form poisonous formaldehyde at temperatures around 40 °C. Aspartame is contraindicated in people with phenylketonuria, a metabolic disorder in which the body is unable to convert phenylalanine to tyrosine. 

Aspartame is the subject of controversy about the possible negative effects associated with its use, such as headaches, fatigue, hearing disorders, anxiety, depression, insomnia, and increased risk of cerebrovascular events and cancer [161,162]. The direct association between higher aspartame consumption and increased incidence of cerebrovascular events and cancer has been confirmed by the findings of the large-scale NutriNet-Santé population-based prospective cohort study published in 2022 [163,164]. Hence, although there was no convincing evidence from experimental or human data that aspartame has adverse effects after ingestion, the cancer research arm of the World Health Organization (WHO) has classified aspartame as ‘possibly carcinogenic’ [165]. 

The pathogenesis of the above-mentioned risks is not clear. Adverse effects are apparently due to the effects of phenylalanine, L-Asp, and methanol that arise during aspartame degradation in the gut [161,162]. Increased levels of L-Asp and phenylalanine, which cross the blood–brain barrier, can play a role in changes in brain neurochemistry. Excess L-Asp can disrupt NMDA receptor signaling and be converted by cAST into glutamate, which is linked to the hyperexcitability of neurons. Excess of phenylalanine impairs the transport of tyrosine and tryptophan through the blood–brain barrier, inhibits the flux through tyrosine hydroxylase and tryptophan hydroxylase, and inhibits the synthesis of dopamine, norepinephrine, and serotonin [162]. Methanol is highly toxic, blindness is the best-known consequence of its ingestion, and it is potentially carcinogenic because it is metabolized into formic acid, which can damage DNA [166]. Nevertheless, it is supposed that its role in carcinogenesis is unlikely since the amount of methanol generated by aspartame breakdown is negligible [161,165].

Several animal studies have shown that prenatal exposure to low-calorie sweeteners, including aspartame, leads to higher selection and taste preference for sweet foods and elevated fasting blood glucose, as well as reduced insulin sensitivity in later life [167]. The studies performed in Canada and Denmark have shown that pregnant women consuming artificial sweeteners had a higher risk of an infant being overweight [168,169]. Therefore, aspartame should be avoided by pregnant and nursing women.

#### 6.2.2. Neotame 

Neotame (N-(3,3-dimethylbutyl)-L-alpha-aspartyl-L-phenylalanine) is a high-intensity artificial sweetener. The sweetness potency of neotame is approximately 6000 to 10,000 times higher than that of sucrose [160]. Unlike aspartame, it is stable at high temperatures. It is used in the production of cookies, yogurts, carbonated drinks, and chewing gums [170]. Studies describing the health risks reported with aspartame have not yet been published.

### 6.3. D-Aspartate

The studies reporting that D-Asp activates the hypothalamic–pituitary–gonadal axis and stimulates the synthesis and release of testosterone have suggested its use as a testosterone booster for infertile men and to increase muscle mass and strength of athletes and bodybuilders [51,89]. Hence, D-Asp emerged on the market as a supplement claimed to improve erection, male fertility, and muscle performance. However, reports on its effects are not conclusive.

The results of the systematic review of 23 animal and 4 human studies published in 2017 demonstrated that exogenous D-Asp enhanced testosterone levels in animals, whereas human studies yielded inconsistent results [171]. The authors have concluded that the likely cause of the inconsistent results in humans was the low quality of primary studies and that more high-quality randomized controlled trials need to be conducted. The studies evaluating the effects of D-Asp supplementation (3 or 6 g daily for 1 or 3 months) in resistance-trained men did not report a positive influence on training outcomes [172,173].

Unlike a relatively large number of studies evaluating the effect of D-Asp on fertility in males, studies evaluating its role in females are rare. One study has demonstrated that D-Asp is present in human ovarian follicular fluid, and its concentration is higher in younger than in older women [174]. Unfortunately, we found no studies examining the effect of exogenous D-Asp on female fertility.

Because chronic exposure to D-Asp alters glutamatergic transmission and deteriorates hippocampal functions in mice [8], the effects of D-Asp supplementation on testosterone levels, fertility, and muscle performance should be investigated simultaneously with its influence on brain functions.

### 6.4. Asparagine

Asparagine is considered a suitable supplement for improving the metabolism of the intestinal mucosa due to its stimulating effect on ornithine decarboxylase and supply of L-Asp and glutamate by asparaginase reaction [175].

### 6.5. Side Effects of Aspartate and Aspartate-Containing Supplements

Increased intake of any amino acid, including aspartate, can have a variety of adverse effects on the body. Those are mainly increased concentrations of its metabolites, imbalance in amino acid concentrations, and impaired transport of other amino acids through cell membranes due to the competition for a carrier. Hence, although L-Asp administration did not result in any adverse effects in human studies examining its ergogenic effects [176,177,178,179], animal experiments evaluating the effects of exogenous aspartate on the metabolism and resorption of amino acids, especially glutamic acid, and intestinal function and nitrogen balance are necessary. Research is also required to examine how increased intake of aspartame and other artificial sweeteners in pregnancy affects maternal and child health and its potential influence on carcinogenesis. 

## 7. Conclusions

It is concluded that L-Asp has a crucial role in mitochondrial function and linking the reactions of amino acid catabolism, PNC, glycolysis, gluconeogenesis, protein synthesis, and cell proliferation. Due to the essential role of L-Asp in cell proliferation and reports of increased risk of cancer in subjects consuming higher amounts of aspartame, further research needs to examine strategies of targeting L-Asp metabolism to fight cancer and the influence of increased L-Asp consumption on carcinogenesis, respectively. Well-designed intervention-controlled trials are necessary to evaluate the effects of increased aspartame intake during pregnancy. The recent advances in the role of D-Asp in brain neurochemistry warrant further studies on its unique significance in brain development and neurotransmission and therapeutic potential in psychiatric and neurologic disorders.

## Figures and Tables

**Figure 1 nutrients-15-04023-f001:**
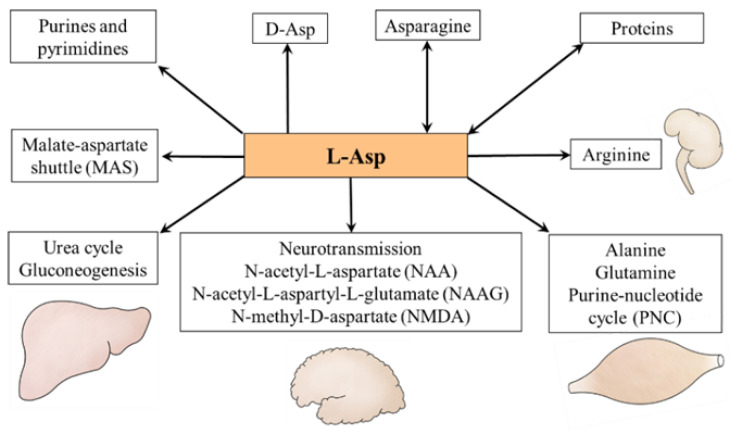
**Schematic outline of the physiologic importance of L-Asp**. Some metabolic pathways are tissue-specific.

**Figure 2 nutrients-15-04023-f002:**
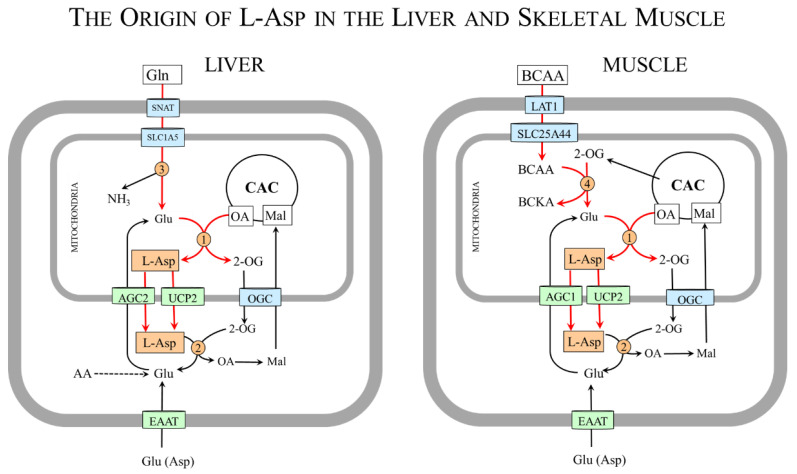
**The origin of L-Asp in the liver and skeletal muscle.** The main source of L-Asp is its synthesis from oxaloacetate and glutamate by mAST. The source of oxaloacetate is the citric acid cycle, and the main source of glutamate is glutamine in the liver, whereas the BCAAs are the main donor of amino groups for glutamate synthesis in muscles. The reactions showing the role of glutamine and BCAA in L-Asp synthesis are shown by red arrows. 1, mAST; 2, cAST; 3, glutaminase; and 4, BCAA aminotransferase. Abbreviations: AA, amino acids; AGC, aspartate–glutamate carrier; BCAA, branched-chain amino acids; BCKA, branched-chain keto acids; CAC, citric acid cycle; EAAT, excitatory amino acid transporter; LAT1, large neutral amino acid transporter 1; Mal, malate; OA, oxaloacetate; OGC, 2-oxoglutarate carrier; SNAT, sodium-neutral amino acid transporter; UCP2, uncoupling protein 2; and 2-OG, 2-oxoglutarate.

**Figure 3 nutrients-15-04023-f003:**
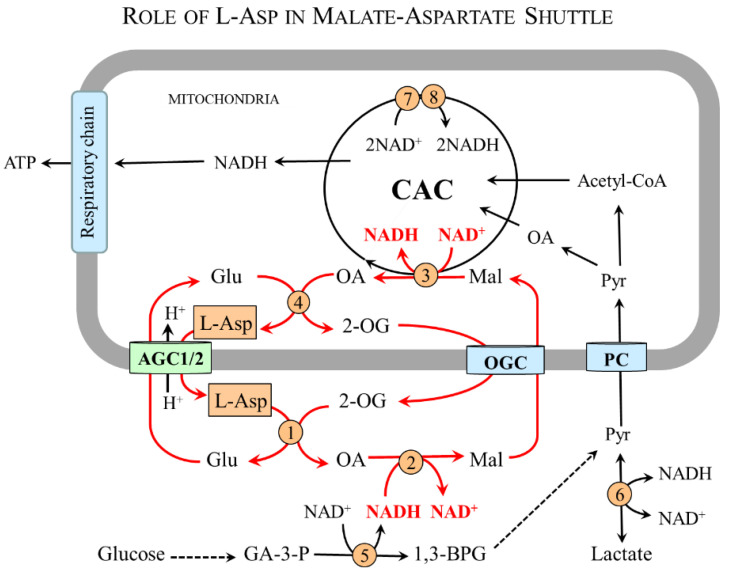
**MAS and its relationship to glycolysis and citric acid cycle.** The MAS consists of AGC1/2 and 2-oxoglutarate carrier (OGC) and two enzymes, malate dehydrogenase (MDH) and AST, found both in the mitochondria and the cytosol. The reactions of MAS are shown by red arrows. 1, cAST; 2, cMDH; 3, mMDH; 4, mAST; 5, 3-phosphoglyceraldehyde dehydrogenase; 6, lactate dehydrogenase; 7, isocitrate dehydrogenase; and 8, 2-OG dehydrogenase complex. Abbreviations: AGC1/2, aspartate–glutamate carrier 1/2; CAC, citric acid cycle; GA-3-P, glyceraldehyde-3-phosphate; Mal, malate; OA, oxaloacetate; OGC, 2-oxoglutarate carrier; PC, pyruvate carrier; Pyr, pyruvate; 1,3-BPG, 1,3-bisphosphoglycerate; 2-OG, 2-oxoglutarate.

**Figure 4 nutrients-15-04023-f004:**
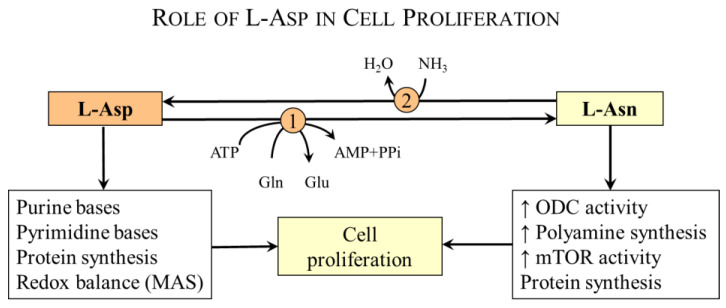
**L-Asp and cell proliferation**. L-Asp is essential for maintaining redox balance and the synthesis of purine/pyrimidine bases, polyamines, proteins, and asparagine. 1, Asparagine synthetase; 2, asparaginase. Abbreviations: ODC, ornithine decarboxylase; mTOR, mammalian target of rapamycin.

**Figure 5 nutrients-15-04023-f005:**
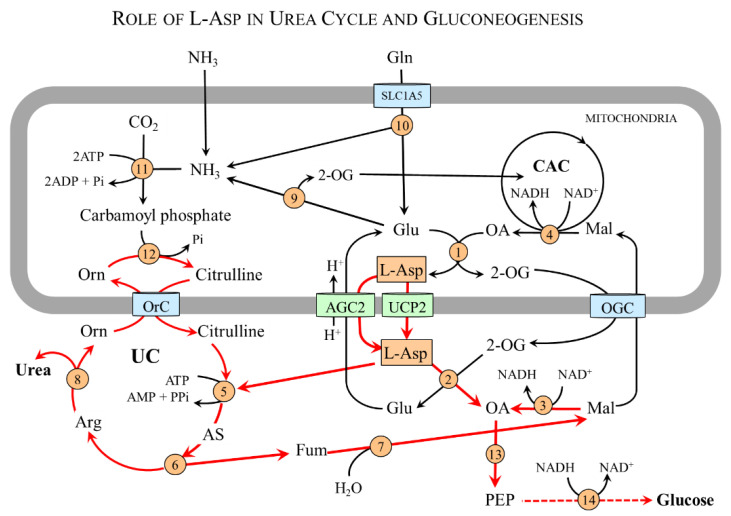
**The role of L-Asp in urea cycle and gluconeogenesis.** L-Asp delivered from the mitochondria to the cytosol can enter gluconeogenesis via oxaloacetate synthesized directly by cAST or through the urea cycle. The reactions showing the role of L-Asp in the urea cycle and gluconeogenesis are shown by red arrows. 1, mAST; 2, cAST; 3, cMDH; 4, mMDH; 5, argininosuccinate synthetase; 6, argininosuccinate lyase; 7, fumarate hydratase; 8, arginase; 9, glutamate dehydrogenase; 10, glutaminase; 11, carbamoyl phosphate synthetase 1; 12, ornithine carbamoyltransferase; 13, phosphoenolpyruvate carboxykinase; and 14, 3-phosphoglyceraldehyde dehydrogenase. Abbreviations: AGC2, aspartate–glutamate carrier 2; AS, argininosuccinate; CAC, citric acid cycle; Fum, fumarate; Mal, malate; OA, oxaloacetate; OGC, 2-oxoglutarate carrier; OrC, ornithine/citrulline carrier; PEP, phosphoenolpyruvate; Pi, inorganic phosphate; UC, urea cycle; UCP2, uncoupling protein 2; and 2-OG, 2-oxoglutarate.

**Figure 6 nutrients-15-04023-f006:**
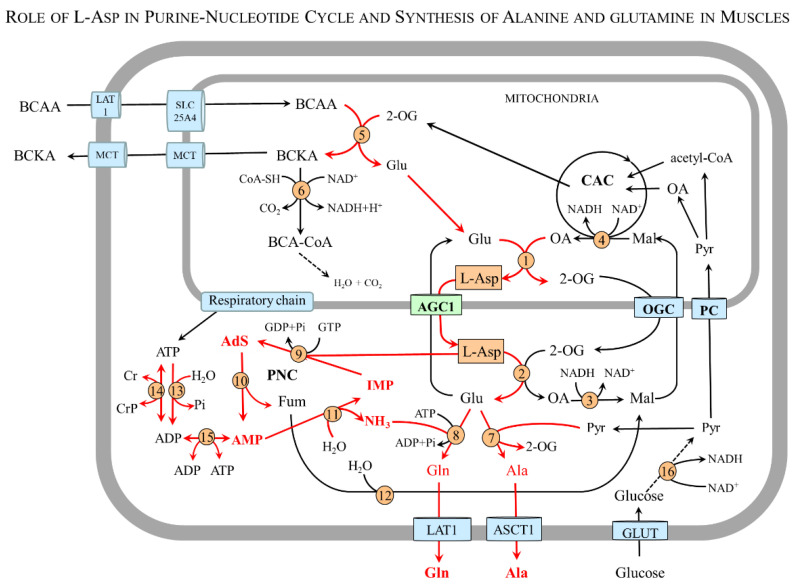
**Putative role of L-Asp in PNC and alanine and glutamine synthesis in muscles.** The main pathways are shown by red arrows. 1, mAST; 2, cAST; 3, cMDH; 4, mMDH; 5, BCAA aminotransferase; 6, BCKA dehydrogenase; 7, ALT; 8, glutamine synthetase; 9, adenylosuccinate synthetase; 10, adenylosuccinate lyase; 11, adenylate deaminase; 12, fumarase; 13, ATPases; 14, creatine kinase; 15, adenylate kinase (myokinase); and 16, 3-phosphoglyceraldehyde dehydrogenase. Abbreviations: AdS, adenylosuccinate; AGC1, aspartate–glutamate carrier 1; ASCT1, alanine, serine, cysteine, and threonine carrier 1; BCAA, branched-chain amino acids; BCA-CoA, branched-chain acyl-CoA; BCKA, branched-chain keto acids; CAC, citric acid cycle; Cr, creatine; CrP, creatine phosphate; Fum, fumarate; GLUT, glucose transporter; IMP, inosine monophosphate; LAT1, large neutral amino acid transporter 1; Mal, malate; MCT, monocarboxylate transporter; OA, oxaloacetate; OGC, 2-oxoglutarate carrier; PC, pyruvate carrier; Pi, inorganic phosphate; PNC, purine-nucleotide cycle; Pyr, pyruvate; and 2-OG, 2-oxoglutarate.

**Figure 7 nutrients-15-04023-f007:**
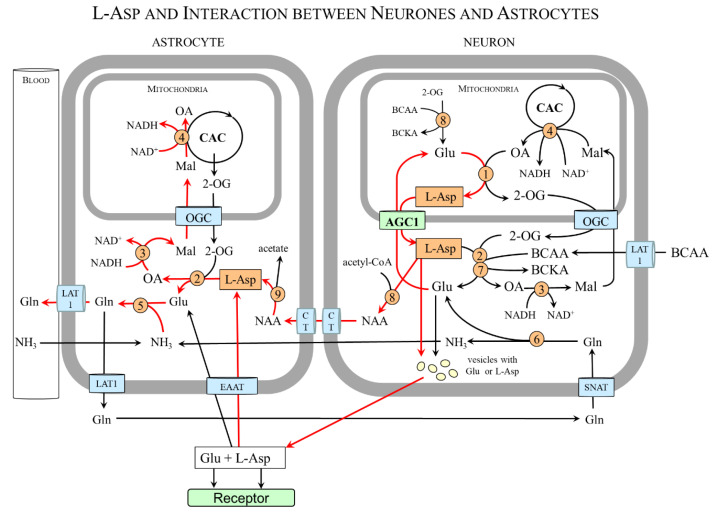
**L-Asp and interaction between neurons and astrocytes**. L-Asp released from neurons ensures glycolysis, redox balance, and ammonia detoxification to glutamine in astrocytes. The main pathways are shown by red arrows. 1, mAST; 2, cAST; 3, cMDH; 4, mMDH; 5, glutamine synthetase; 6, glutaminase; 7, BCAA aminotransferase; 8, aspartate N-acyltransferase; and 9, aspartoacylase. Abbreviations: AGC1, aspartate–glutamate carrier 1; BCAA, branched-chain amino acids; BCKA, branched-chain keto acids; CAC, citric acid cycle; CT, carboxylate transporter; EAAT, excitatory amino acid transporter; LAT1, large neutral amino acid transporter 1; Mal, malate; NAA, N-acetyl-L-aspartate; OA, oxaloacetate; OGC, 2-oxoglutarate carrier; SNAT, sodium-neutral amino acid transporter; and 2-OG, 2-oxoglutarate.

**Figure 8 nutrients-15-04023-f008:**
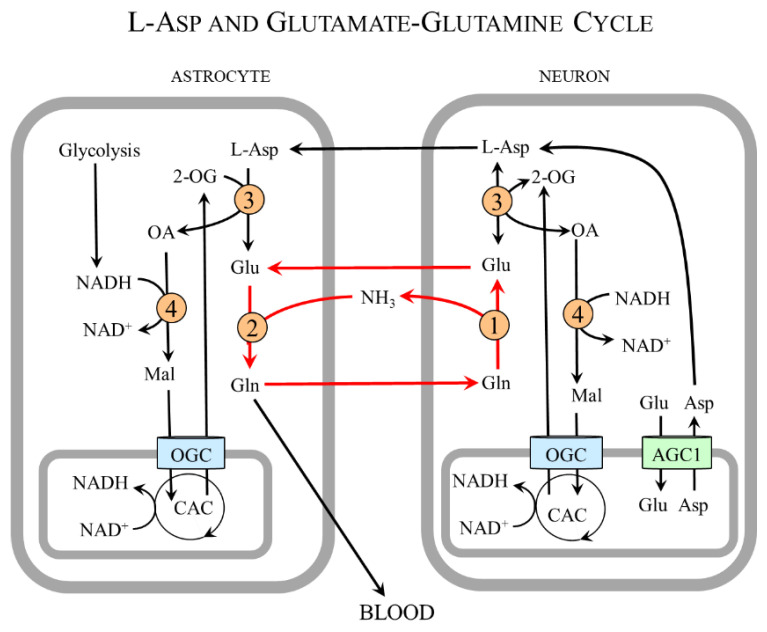
**Schematic of the relationship of L-Asp and glutamate–glutamine cycle.** L-Asp acquired by astrocytes from neurons connects to the glutamate–glutamine cycle via glutamate formed by cAST. The glutamate–glutamine cycle is shown by red arrows. 1, glutaminase; 2, glutamine synthetase; 3, cAST; 4, cMDH. Abbreviations: CAC, citric acid cycle; Mal, malate; OA, oxaloacetate; OGC, 2-oxoglutarate carrier; and 2-OG, 2-oxoglutarate.

**Figure 9 nutrients-15-04023-f009:**
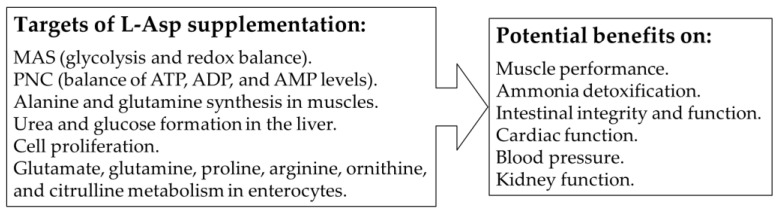
**Predicted benefits of L-Asp supplementation**.

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
