# Peer review of "Aspartic Acid in Health and Disease"

_nutrients, 2023, doi:10.3390/nu15184023_

Round 1

Reviewer 1 Report

The review by Holecek entitled “Aspartic Acid in Health and Disease” gives a comprehensive overview of the role of this amino acid in metabolism, physiology and pathology. This reviewer has only minor comment for improving the manuscript before acceptance and publication.

Minor comments

- Line 55: “the second dicarboxylic amino acid” should be “the other dicarboxylic amino acid”.

- Section 2.1 or 3.1: it should be more clearly described how the low amounts of Asp (or digested peptides containing Asp) are thought to be absorbed in the gastrointestinal tract. Are there specific transporters or the same as those that import Asp in all the other cell types of the body?

- Section 2.2: it should be mentioned that also UCP5 has the ability to transport L-Asp.

- Lines 129-132 and Fig. 2 (left): maybe it should be mentioned that cytosolic glutamate can be imported into mitochondria by SLC25A18 or SLC25A22.

- Lines 178-179: “A single-letter symbol "D" is used to indicate aspartate presence in protein molecules which are around 8%” should be something like “The single-letter amino acid symbol "D" is used in protein sequences to indicate the presence of aspartate, which global frequency is around 8%”.

- Lines 201-202: “The NADH is funnelled into the respiratory chain” should be “The electrons of NADH are funnelled into the respiratory chain”.

- Line 245: “in textbooks biochemistry” should be “in biochemistry textbooks”.

- Line 297: “releases L-Asp skeleton as fumarate” should be “releases L-Asp carbon skeleton as fumarate”.

- Fig. 5. The arrow pointing towards Urea in reaction 8 (low left part of the figure) almost appears to indicate a reverse reaction. It should be pointed so that it is clear that Orn and Urea are derived from Arg.

- Line 380: “13, ATPase” should be “13, ATPases”.

- Section 5.1: it would be suitable to specify the diseases with their OMIM numbers.

Author Response

Dear Reviewer No. 1,

Thank you very much for your comments, which helped us to revise the manuscript. We believe that the contents and the clarity of our paper are much improved in the revised version. Below are point-by-point our responses.

Milan Holeček

Comment

 - Line 55: “the second dicarboxylic amino acid” should be “the other dicarboxylic amino acid”.

 Response

Done. Thank you.

Comment

- Section 2.1 or 3.1: it should be more clearly described how the low amounts of Asp (or digested peptides containing Asp) are thought to be absorbed in the gastrointestinal tract. Are there specific transporters or the same as those that import Asp in all the other cell types of the body?

Response

In the revised version is stated:”… most L-Asp provided from food is transported through the apical membrane of enterocytes by SLC1A1 (EAAT3)...”  Please see section 3.1., first two lines.

Comment

- Section 2.2: it should be mentioned that also UCP5 has the ability to transport L-Asp.

Response

In the revised version of the manuscript is stated: “It has recently been discovered that the ability to transport L-Asp has also UCP5 (BMCP1, brain mitochondrial carrier protein 1) encoded by SLC25A14 [35]).” 

Comment

- Lines 129-132 and Fig. 2 (left): maybe it should be mentioned that cytosolic glutamate can be imported into mitochondria by SLC25A18 or SLC25A22.

Response

Thank you. The sentence has been edited as follows: “Under ordinary conditions, the net flux through mAST is toward the direction of L-Asp because of a continuous supply of glutamate from the cytosol into the mitochondria by several transporters including SLC25A12 (AGC1), SLC25A13 (AGC2), SLC25A18, and SLC25A22,...”. Please see section 3.1.

Comment

- Lines 178-179: “A single-letter symbol "D" is used to indicate aspartate presence in protein molecules which are around 8%” should be something like “The single-letter amino acid symbol "D" is used in protein sequences to indicate the presence of aspartate, which global frequency is around 8%”.

Response

Edited as suggested by the reviewer. Please see section 4.1. Thank you.

Comment

- Lines 201-202: “The NADH is funnelled into the respiratory chain” should be “The electrons of NADH are funnelled into the respiratory chain”.

Response

Edited as suggested by the reviewer. Please see section 4.2. Thank you.

Comment

- Line 245: “in textbooks biochemistry” should be “in biochemistry textbooks”.

 Response

Done. Thank you.

- Comment

Line 297: “releases L-Asp skeleton as fumarate” should be “releases L-Asp carbon skeleton as fumarate”.

Response

Edited. Thank you. Please see section 4.5.1.

Comment

- Fig. 5. The arrow pointing towards Urea in reaction 8 (low left part of the figure) almost appears to indicate a reverse reaction. It should be pointed so that it is clear that Orn and Urea are derived from Arg.

Response

Edited. Thank you.

Comment

- Line 380: “13, ATPase” should be “13, ATPases”.

Response

Edited. Thanks.

Comment

- Section 5.1: it would be suitable to specify the diseases with their OMIM numbers.

Response

Done. Please see the revised version.

Reviewer 2 Report

Asp plays an important role in physiological metabolic activity. This manuscript reviews the origins and pathways of aspartate metabolism in physiological and various pathological conditions, the therapeutic potential of targeting aspartate metabolism pathways, and the use of aspartate and aspartate-containing substances as nutritional supplements.

Although this is a popular subject with high research value and falls within the scope of the journal, there are certain issues that the authors need to address specifically to improve the quality of the submission.

For my part, as a review, this article contains too much description in terms of scientific introduction and too little in terms of summarizing research progress, pointing out research deficiencies or controversies, and presenting personal perspectives.

 There were too many keywords that needed to be streamlined.

The logic of the introduction section needs to be optimized. Authors are recommended to give a brief summary of research progress and point out research gaps.

I am personally more interested in Sections 2.2 and 6.  Currently, studies related to amino acid and mitochondrial metabolism and daily dietary supplementation are also quite popular. If possible, the author is advised to plot the mechanism figures of the two parts to help the reader better understand the relevant content.

As discussed in Section 6.3, due to the role of aspartic acid in the regulation of testosterone, some studies have suggested that D-Asp supplementation may be associated with increased fertility, but most studies on the relationship between D-Asp and fertility have focused on male fertility and are controversial. 

Are there any studies on D-ASP supplementation and female fertility?

Are the findings controversial?

What are the reasons for the different results?

What does the author think of these controversies?

It is recommended that the authors supplement this section.

To the best of my knowledge, taking amino acid supplements, including aspartate, can have a variety of adverse effects on the body. This is because the supplement increases the intake of one amino acid, which can lead to an imbalance in the body's nitrogen balance. A negative nitrogen balance means that the amount of nitrogen leaving the body through urine is higher than the amount of nitrogen entering the body through the mouth. This can lead to anemia, reduced resistance, impaired metabolism and fatty liver development. Generally, aspartic acid supplements are not recommended for pregnant or nursing women. Please revise and improve Section 6.5 and the conclusions of the manuscript.

There are also minor issues such as typos, such as an extra period in lines 219, 457, 472. Please check for similar errors throughout the manuscript.

In summary, I recommend that the article be further evaluated after major revision.

Fine.

Author Response

Dear Reviewer No. 2,

Thank you very much for your comments, which helped us to revise the manuscript. We believe that the contents and the clarity of our paper are much improved in the revised version. Below are point-by-point our responses.

Milan Holeček

Comment

For my part, as a review, this article contains too much description in terms of scientific introduction and too little in terms of summarizing research progress, pointing out research deficiencies or controversies, and presenting personal perspectives.

Response

We have included notes related to research deficiencies or controversies and our personal opinions in the manuscript.  Some are based on your comments. Here are examples:

Section 6.2.1.: “Therefore, aspartame should be avoided by pregnant and nursing women.”

Section 6.3.: “Unfortunately, we found no studies examining the effect of exogenous D-Asp on female fertility.”

Section 6.3.:  “…the effects of D-Asp supplementation on testosterone levels, fertility, and muscle performance should be investigated simultaneously with its influence on brain functions.”

Section 6.5.: “…animal experiments evaluating the effects of exogenous aspartate on the metabolism and resorption of amino acids, especially glutamic acid, and intestinal function and nitrogen balance are necessary.”

Section 6.5.: “Research is also required to examine how increased intake of aspartame and other artificial sweeteners in pregnancy affects maternal and child health and its potential influence on carcinogenesis.”

Comment

The logic of the introduction section needs to be optimized. Authors are recommended to give a brief summary of research progress and point out research gaps.

Response

I accept the reviewer's opinion that the Introduction of the previous version of the manuscript is not well written. Therefore I have replaced the unattractive paragraph "Recent investigations..." with the following sentence:  "The focus is on the role of mitochondrial carriers in the compartmentalization of L-Asp and glutamate metabolic pathways between mitochondria and cytosol [2-5] and new insights into the role of aspartate in cell-to-cell interactions and neurotransmission in the brain, cell proliferation, glycolysis, gluconeogenesis, diabetes, psychiatric and neurologic disorders, liver cirrhosis, and cancer [6-18].”  I believe that this edit of the Introduction section can attract more interest than the previous version.

In my opinion, there is not enough space to summarize progress and point out research gaps in the Introduction section because, in this article, aspartic acid is discussed in a wide range of areas. 

Comment

I am personally more interested in Sections 2.2 and 6.  Currently, studies related to amino acid and mitochondrial metabolism and daily dietary supplementation are also quite popular. If possible, the author is advised to plot the mechanism figures of the two parts to help the reader better understand the relevant content.

Response

The following sentence and Figure have been added to section 6:

“Recommendations of L-Asp as a dietary supplement are based on its importance for mitochondrial function, the stimulating effect on PNC and alanine and glutamine synthesis in muscles, urea and glucose formation in the liver, and cell proliferation and its close relationship to the metabolism of glutamate, glutamine, proline, arginine, ornithine, and citrulline (Figure 9).

Figure 9.  Predicted benefits of L-Asp supplementation.

Comment

As discussed in Section 6.3, due to the role of aspartic acid in the regulation of testosterone, some studies have suggested that D-Asp supplementation may be associated with increased fertility, but most studies on the relationship between D-Asp and fertility have focused on male fertility and are controversial.

Are there any studies on D-ASP supplementation and female fertility?

Are the findings controversial?

What are the reasons for the different results?

What does the author think of these controversies?

It is recommended that the authors supplement this section.

Response

The section on D-Asp as a possible nutritional supplement has been rewritten. I believe I have responded satisfactorily to the reviewer's constructive comments. Please see section 6.3. 

Comment

To the best of my knowledge, taking amino acid supplements, including aspartate, can have a variety of adverse effects on the body. This is because the supplement increases the intake of one amino acid, which can lead to an imbalance in the body's nitrogen balance. A negative nitrogen balance means that the amount of nitrogen leaving the body through urine is higher than the amount of nitrogen entering the body through the mouth. This can lead to anemia, reduced resistance, impaired metabolism and fatty liver development. Generally, aspartic acid supplements are not recommended for pregnant or nursing women. Please revise and improve Section 6.5 and the conclusions of the manuscript.

Response

I thank the reviewer for the suggestions. I have edited the article as follows:

(i)   At the end of section 6.2.1 is stated: “Several animal studies have shown that prenatal exposure to low-calorie sweeteners including aspartame leads to higher selection and taste preference for sweet foods and elevated fasting blood glucose as well as reduced insulin sensitivity in later life [167]. The studies performed in Canada and Denmark have shown that pregnant women consuming artificial sweeteners had a higher risk of an infant being overweight [168,169]. Therefore, aspartame should be avoided by pregnant and nursing women.”

(ii)   Section 6.5 has been rewritten. At the beginning of it is stated: “Increased intake of any amino acid including aspartate can have a variety of adverse effects on the body. Those are mainly increased concentrations of its metabolites, imbalance in amino acid concentrations, and impaired transport of other amino acids through cell membranes due to the competition for a carrier.”

(iii)  At the end of section 6.5 is stated: ”Research is also required to examine how increased intake of aspartame and other artificial sweeteners in pregnancy affects maternal and child health and its potential influence on carcinogenesis.”

Comment

There are also minor issues such as typos, such as an extra period in lines 219, 457, 472. Please check for similar errors throughout the manuscript.

Response

Done. Thank you.

Comment

Comments on the Quality of English Language

Fine.

Response

Thanks.

Round 2

Reviewer 2 Report

  • All responses except for the keyword issue are acceptable.

  • Fine.